# Method of Moments Optimization of Distributed Raman Amplification in Fibers with Randomly Varying Birefringence

**Antonio Ortiz-Mora †, Pedro Rodríguez †, Antonio Díaz-Soriano *,† , David Martínez-Muñoz † and Antonio Dengra †**

Physics Department, University of Córdoba, 14014 Andalucia, Spain; fa2ormoa@uco.es (A.O.-M.); pm1rogap@uco.es (P.R.); f12mamud@uco.es (D.M.-M.); fa1desaa@uco.es (A.D.)
* Correspondence: f62disoa@uco.es; Tel.: +34-957-212-551
† These authors contributed equally to this work.

**Abstract:** In this work, we develop a vectorial model, for optical communications, using distributed Raman amplification (DRA) applied to vectorial soliton pulses in monomode fiber optics. The main result, a dependent polarization effective Raman gain coefficient, is calculated considering the random birefringence character of the fiber and the relative mismatch between the continuous wave (CW) pump and the signal. The Method of Moments allows the determination of the optimal initial conditions to achieve a better performance in our system. With these starting values, the simulations carried out with the Split-Step Fourier Method (SSFM) elucidate the influence of the peak power and prechirping initial phase control on the vectorial soliton propagation.

**Keywords:** Raman amplification; vectorial soliton; optical communications; random birefringence

## 1. Introduction

Nonlinear effects in optical fibers have a clear influence in signal propagation in a variety of interesting phenomena that can disrupt the transmission of the optical signal. One of them is stimulated Raman scattering (SRS), which has allowed the development of distributed Raman amplification systems (DRA).

Historically, the investigation of these systems began a few decades ago and a formal theoretical analysis of the SRS phenomenon in birefringent single-mode optical fibers was clearly substantiated [1,2], the study of the coupling between pump and the Stokes signal wave amplitudes was obtained and the gain for the different polarization components without taking into account dispersive effects was identified.

Later on, a vector theory of the SRS process that describes the polarization effects in Raman fiber amplifiers (RFA) was developed [3], the results showed how polarization mode dispersion (PMD) caused fluctuations in the continuous wave signal when it propagates through the amplifying device. This model was later refined [4] to describe the vector interaction between strong counterpropagation pumping and a weak Stokes signal in birefringent fibers.

In order to complete the model, a modification of the technique of averaging over the length of random birefringence correlation in the fibers to study the properties of polarization in RFA is proposed [5–7], along with the works in [8–10] which analyze the interaction of two counter-propagating polarized beams in a randomly birefringent fiber via the Kerr and Raman effects.

Finally, a direct stochastic model to describe the interaction between Raman scattering in fibers and their random birefringence was developed [11]. However, none of these works pay attention

to the combination of Raman amplification with the dispersive phenomena and those of phase self-modulation that arise in the fiber when short soliton pulses are propagated.

Therefore, the objective of this work is to complete a vector model of Raman amplification applied to short solitonic pulses. To accomplish this task, a significant number of new model assumptions are used, e.g., the random fiber birefringence properties are averaged over long enough runs, the relative coupling between the pumping and signal polarization states is taken into account, the effective average gain conditions recorded by a pulsed soliton signal are calculated and the optimal soliton launch conditions using the method of moments [12] determined.

The organization of the paper is as follows. In Section 2, the theoretical model is explained; in Section 3, the results of the simulation, using the Method of the Moments for the initial conditions and the SSFM method [13], of the coupled equations obtained in the previous section are presented, along with the influence of the peak power and the initial phase control on these; in Section 4, we make an interpretation of the aforementioned results.

## 2. Theoretical Model

This section has three subsections. In the first one, the model of a vectorial SRS system is explained, the Raman gain factor in random birefringence fibers is calculated in the second subsection and we end up using the previous results of the section and making an average over all the orientations, due to the aleatory nature of the fiber, which gives us the propagation equations of our system in the final part.

### 2.1. Basic Model for the Vectorial Soliton Raman Amplification

Our study of random birefringence influence in soliton Raman amplification process starts with the electric field envelope scalar-propagation equation, which represents an optical pulse under the effect of Stimulated Raman scattering (SRS) [14], including the orthogonal polarization components [15,16]. A more general case, a fiber with elliptical birefringence, is considered, where the unit vectors of the principal axes in a section of the birefringent fiber are $\hat{x}$ and $\hat{y}$, the electric field associated with an optical wave signal of frequency $\omega_s$ and arbitrary polarization that copropagates with a pumping $\omega_p$, from which it receives gain through *SRS*, then:

$$\mathbf{E}(\mathbf{r}, t) = \frac{1}{2} \sum_{j=p,s} \left[ \hat{\mathbf{e}}_{\mathbf{x}} E_{j,x}(\mathbf{r}, t) + \hat{\mathbf{e}}_{\mathbf{y}} E_{j,y}(\mathbf{r}, t) \right] + c.c., \tag{1}$$

here, $E_{j,x}$ and $E_{j,y}$ represent the polarization components complex amplitudes of signal and pump field with $j = s, p$, and the vectors $\hat{\mathbf{e}}_{\mathbf{x}}$ and $\hat{\mathbf{e}}_{\mathbf{y}}$ are the orthonormal elliptic polarization eigenvectors related to the vectors $\hat{x}$ and $\hat{y}$ in the form:

$$\hat{\mathbf{e}}_{\mathbf{x}} = \frac{\hat{x} + ir\hat{y}}{\sqrt{1 + r^2}}, \qquad \hat{\mathbf{e}}_{\mathbf{y}} = \frac{r\hat{x} - i\hat{y}}{\sqrt{1 + r^2}} \tag{2}$$

in addition, the fiber core ellipcity degree is represented by the $r$ parameter that occurs due to unintended kinks during the fiber manufacturing process. The $\theta$ angle is usually introduced for its characterization in the form:

$$r = tan(\theta/2) \tag{3}$$

We assume the almost transverse character of the fields, so the axial components $E_{j,z}$ are supposed to remain small enough and can be neglected. The non-linear signal polarization components $P_{s,j}^{NL}$, where the third-order non-linear response tensor appears explicitly, including both the electronic and

vibrational contributions of the medium, with $j = x, y$; $m = x, y$ and $j \neq m$ can be obtained in the form [17]:

$$
\begin{aligned}
P_{s,j}^{NL}(\mathbf{r},t) = \frac{3\epsilon_0}{4}\chi_{xxxx}^{(3)}\Big\{ & \left(|E_{s,j}|^2 + B|E_{s,m}|^2\right)E_{s,j} + C(E_{s,j}^* E_{s,m})E_{s,m} + \\
& + D\left[(E_{s,m}^* E_{s,j})E_{s,j} + \left(|E_{s,m}|^2 + 2|E_{s,j}|^2\right)E_{s,m}\right] + \\
& + (2 - f_R)\left(|E_{p,j}|^2 + |E_{p,m}|^2\right)E_{s,j} + \\
& + \frac{f_R}{2}\left[2|E_{p,j}|^2 Im\left[\widetilde{R_a}(\Omega_R) + \widetilde{R_b}(\Omega_R)\right] + \\
& + |E_{p,m}|^2 Im\left[\widetilde{R_b}(\Omega_R)\right]\right]e^{-i(\omega_m - \omega_j)t}E_{s,j}\Big\}
\end{aligned}
\tag{4}
$$

where $E_{s,j} = E_{s,j}(\mathbf{r},t)$ represents the signal electric field and $E_{p,j} = E_{p,j}(\mathbf{r},t)$ accounts for the pumping. We have also explicitly introduced the isotropic $\widetilde{R_a}(\Omega_R)$ and the anisotropic part $\widetilde{R_b}(\Omega_R)$ of the Raman response that allows a better modelling of the pump–signal interaction [18,19]. The parameters $B$, $C$ and $D$ are related to the ellipticity angle $\theta$ in the form:

$$
B = \frac{2 + 2\sin^2\theta}{2 + \cos^2\theta}, \quad C = \frac{\cos^2\theta}{2 + \cos^2\theta}, \quad D = \frac{\sin\theta\cos\theta}{2 + \cos^2\theta}
\tag{5}
$$

The factors appearing on the right side of (4) have a clear physical origin. On the one hand, the first summation shows the self-phase modulation (SPM) contribution due to the same polarization component, while on the other hand, the cross-phase modulation effects (XPM), controlled by the fiber birefringence become a non-linear phase shift. Furthermore, given the presence of the pumping field, the terms due to the non-linear phase coupling between its components with those of the signal also appear.

Finally, the rest of the addends account for the effect of the SRS that provides gain from the coupling of the signal field components with those of the co-polarized and orthogonally polarized pumping.

Later on, the two signal polarization components propagation equations along the single-mode fiber can be factored into the form [13,17]:

$$
E_{s,j}(\mathbf{r},t) = \mathbb{H}_{s,j}(x,y)A_{s,j}(z,t)\exp(i\beta_{s,j}z)
\tag{6}
$$

with $j = x, y$, where $\mathbb{H}_{s,j}(x,y)$ accounts for the transverse signal spatial distribution mode supported by the fiber, $A_{s,j}(z,t)$ is the slow variation amplitude and $\beta_{s,j}$ is the corresponding propagation constant.

To complete the theoretical description, we consider:

- Dispersive effects, which are included by developing the propagation constant in frequency powers series.
- Losses not dependent on the polarization.
- No-pump depletion is taken into account in CW.

Pumping in CW will produce a cross-modulation phase shift. This will be constant for both slowly varying signal amplitudes $A_{sx}(z,t)$ and $A_{sy}(z,t)$, therefore these components will satisfy the following equations [13]:

$$
\begin{aligned}
\frac{\partial A_{s,x}}{\partial z} + \beta_{1s,x}\frac{\partial A_{s,x}}{\partial t} &+ \frac{i}{2}\beta_{2s}\frac{\partial^2 A_{s,x}}{\partial t^2} + \frac{\alpha}{2}A_{s,x} = \\
&= i\gamma_s\left[\left(|A_{s,x}|^2 + B|A_{s,y}|^2\right)A_{s,x} + CA_{s,x}^* A_{s,y}^2 e^{-2i\Delta\beta_s z}\right] + \\
&+ i\gamma_s D\left[A_{s,y}^* A_{s,x}^2 e^{i\Delta\beta_s z} + \left(|A_{s,y}|^2 + 2|A_{s,x}|^2\right)A_{s,y}e^{-i\Delta\beta_s z}\right] + \\
&+ \frac{g_{R\parallel}(\Omega_R)}{2}P_x A_{s,x} + \frac{g_{R\perp}(\Omega_R)}{2}P_y A_{s,x},
\end{aligned}
\tag{7}
$$

$$
\begin{aligned}
\frac{\partial A_{s,y}}{\partial z} &+ \beta_{1s,y}\frac{\partial A_{s,x}}{\partial t} + \frac{i}{2}\beta_{2s}\frac{\partial^2 A_{s,y}}{\partial t^2} + \frac{\alpha}{2}A_{s,y} = \\
&= i\gamma_s\left[\left(|A_{s,y}|^2 + B|A_{s,x}|^2\right)A_{s,y} + CA_{s,y}^*A_{s,x}^2 e^{2i\Delta\beta_s z}\right] + \\
&+ i\gamma_s D\left[A_{s,x}^*A_{s,y}^2 e^{-i\Delta\beta_s z} + \left(|A_{s,x}|^2 + 2|A_{s,y}|^2\right)A_{s,x}e^{i\Delta\beta_s z}\right] + \\
&+ \frac{g_{R\perp}(\Omega_R)}{2}P_x A_{s,y} + \frac{g_{R\parallel}(\Omega_R)}{2}P_y A_{s,y}
\end{aligned}
\tag{8}
$$

where $P_j (j = x, y)$ represents the pumping polarization components and:

$$
\Delta\beta_s = \beta_x - \beta_y = \frac{2\pi}{\lambda}B_m = \frac{2\pi}{l_B}
\tag{9}
$$

it is related to the fiber modal birefringence, which leads to different group velocities for the two polarization components, given that $\beta_{1s,x} \neq \beta_{1s,y}$. However, the second-order dispersion parameter $\beta_{2s}$ and the non-linear parameter $\gamma_s$ will be the same for both polarization components since they have the same wavelength.

In addition, the contributions of the isotropic and anisotropic molecular response are included so that for pumps copolarized with the signal we have [20]:

$$
g_{R\parallel}(\Omega) = g_a(\Omega) + g_b(\Omega)
\tag{10}
$$

where $g_a = 2\gamma f_R Im\left\{\tilde{R}_a(\Omega)\right\}$ and $g_b = 2\gamma f_R Im\left\{\tilde{R}_b(\Omega)\right\}$, where $f_R$ is the fractional contribution of the delayed Raman response to nonlinear polarization [21]. In the same way, for orthogonally polarized pumping and signal, we have:

$$
g_{R\perp}(\Omega) = \frac{g_b(\Omega)}{2}
\tag{11}
$$

We also add the fiber modal birefrengence-dependent terms, which come from a coherent coupling between the two polarization components and lead to degeneracy in the four-wave mixture (FWM). Its importance depends on the fiber length in which phase coupling conditions are satisfied and on the interference with the SRS process, which we minimize as much as possible, taking care that the frequency degeneracy does not overlap the Raman gain bandwidth [22].

In the case of high birefringence fibers, the Equations (7) and (8) can be simplified considerably, i.e., as the beat length $l_B$ is much smaller than typical propagation distances, the exponential factors that include the term $\Delta\beta_s$ will oscillate very quickly but slightly contribute, on average, to the pulses evolution. If these terms are eliminated, the optical pulse propagation equations in fibers with elliptic birefringence in the Raman gain regime, provided by a (CW) pump, become:

$$
\begin{aligned}
\frac{\partial A_{s,x}}{\partial z} &+ \beta_{1s,x}\frac{\partial A_{s,x}}{\partial t} + \frac{i}{2}\beta_{2s}\frac{\partial^2 A_{s,x}}{\partial t^2} + \frac{\alpha}{2}A_{s,x} = \\
&= i\gamma_s\left(|A_{s,x}|^2 + B|A_{s,y}|^2\right)A_{s,x} + \\
&+ \frac{g_{R\parallel}(\Omega_R)}{2}P_x A_{s,x} + \frac{g_{R\perp}(\Omega_R)}{2}P_y A_{s,x},
\end{aligned}
\tag{12}
$$

$$
\begin{aligned}
\frac{\partial A_{s,y}}{\partial z} &+ \beta_{1s,y}\frac{\partial A_{s,x}}{\partial t} + \frac{i}{2}\beta_{2s}\frac{\partial^2 A_{s,y}}{\partial t^2} + \frac{\alpha}{2}A_{s,y} = \\
&= i\gamma_s\left(|A_{s,y}|^2 + B|A_{s,x}|^2\right)A_{s,y} + \\
&+ \frac{g_{R\perp}(\Omega_R)}{2}P_x A_{s,y} + \frac{g_{R\parallel}(\Omega_R)}{2}P_y A_{s,y}
\end{aligned}
\tag{13}
$$

these equations represent an extension of the non-linear scalar propagation equation in Raman gain regime by including the terms associated with polarization.

The coupling parameter $B$ depends on the ellipticity angle $\theta$ and can vary between $2/3$ and $2$ for $\theta$ values between $0$ and $\pi/2$. In the lossless scenario, and if $\theta \approx 35°$, which leads to a value of the parameter $B = 1$, Equations (12) and (13) can be solved using the inverse scattering method [23].

### 2.2. Calculation of the Polarization-Dependent Effective Raman Gain Coefficient in Random Birefringence Fibers

In a fiber that does not maintain polarization because of random birefringence, both the pump and the Stokes waves will experience the same depolarization conditions, thus, having different wavelengths, the relative polarization will be completely random.

If initially both signal and pump are equally polarized, the distance in which the gain decreases will not be the distance in which the initial polarization state of each of them is lost but rather the distance in which the pump and signal polarization states decouple. Beyond this characteristic length, the gain is averaged as $1/2$ of the maximum value in copolarization [24].

To characterize the effective gain coefficient, which takes into account the dependence on polarization, we use the method developed in [25]. We will extend to soliton signals, specifically we will apply in every stage of the birefringent single mode fiber the pumping and Stokes fields:

$$\mathbf{E}_p(\mathbf{r}, t) = \frac{1}{2}\left[\hat{\mathbf{e}}_\mathbf{x} A_{p,x} \mathbb{H}_{p,x}(x, y)e^{ik_{px}z} + \hat{\mathbf{e}}_\mathbf{y} A_{p,y} \mathbb{H}_{p,y}(x, y)e^{ik_{py}z}\right]\exp(-i\omega_p t) + c.c., \tag{14}$$

$$\mathbf{E}_s(\mathbf{r}, t) = \frac{1}{2}\exp\left(\frac{g_{R_b}z}{2}\right)\left[\hat{\mathbf{e}}_\mathbf{x} A_{s,x} \mathbb{H}_{s,x}(x, y)e^{ik_{sx}z} + \hat{\mathbf{e}}_\mathbf{y} A_{s,y} \mathbb{H}_{s,x}(x, y)e^{ik_{sy}z}\right]\exp(-i\omega_s t) + c.c., \tag{15}$$

where $A_{p,x}$, $A_{p,y}$, $A_{s,x}$ and $A_{s,y}$ are complex amplitudes along the principal axes of each fiber section, for pumping and signal. The $k_{px}$, $k_{py}$, $k_{sx}$ and $k_{sy}$ are the corresponding propagation constants. In the case that the scattering would happen in counter-propagation, the following change would be enough

$$k_{sx} \longrightarrow -k_{sx}$$

$$k_{sy} \longrightarrow -k_{sy}$$

and $g_{R_b}$ represents the gain coefficient, we assume that there is no pump depletion.

If these expressions are introduced into the Maxwell field equations, assuming polarized scattering and an instantaneous response in non-linear polarization, the following set of gain-coupled wave equations is obtained:

$$g_{R_b} A_{s,x} = Y|A_{p,x}|^2 A_{s,x} + Y A_{p,x} A_{p,y}^* A_{s,y} K \tag{16}$$

$$g_{R_b} A_{s,y} = Y A_{p,y} A_{p,x}^* A_{s,x} K + Y|A_{p,y}|^2 A_{s,y} \tag{17}$$

where dispersive effects and other non-linear phenomena have not been taken into account.

The $Y$ factor represents the overlap integral of the propagation modes of the fiber, and is inversely proportional to the core effective area [26]. $K$ is a function of the fiber length, its birefringence $\delta n$ and the Raman frequency $\Omega_R$ according to:

$$K = \int_0^L \frac{e^{i\Delta kz}}{L}dz = \exp\left(\frac{i\Delta kL}{2}\right)\frac{sin\left(\frac{\Delta kL}{2}\right)}{\frac{\Delta kL}{2}} \tag{18}$$

finally, $\Delta k$ is related to inverse of the length at which the polarized pump and the Stokes wave are decoupled. In a co-propagation pumping, the following equation would be fulfilled:

$$\Delta k = (k_{px} - k_{py}) - (k_{sx} - k_{sy}) = \frac{\delta n \, \Omega_R}{c} \tag{19}$$

To obtain the gain coefficient in terms of the relative polarization between pump and signal, we consider a CW pump with initial total power $P_0$ linearly polarized fulfilling the condition: $A_{p,x} = A_{p,y}^*$ and $|A_{p,x}|^2 = |A_{p,y}|^2 = P_0/2$, which corresponds to a linear polarization at $\pi/4$. If we substitute in coupled Equations (16) and (17), we obtain the following expression for gain coefficient:

$$g_{R_b}^2 - g_{R_b} Y P_0 + Y^2 \frac{P_0^2}{4} \left(1 - K^2\right) = 0 \qquad (20)$$

with solutions:

$$g_{R_b} = \frac{Y P_0}{2} \left(1 \pm K\right) \qquad (21)$$

here, the plus sign corresponds to a maximum value of the gain dependent on polarization $PDG_{max}$ coefficient, obtained when pumping and signal were co-polarized. On the other hand, the minus sign will give the minimum polarization-dependent gain $PDG_{min}$ that appears in the case of relative orthogonal polarization.

Next, we introduce the typical pump–signal polarizations decoupling length $l_p$. This has to be understood as the distance from which both polarization states appear uncorrelated due to the fiber birefringent properties. To do that, we make use of the statistical model developed, applied for pumping and continuous wave signals in a Raman Fiber Amplifier (RFA) [3]. This distance is evaluated from the PMD coefficient $D_p$:

$$l_p = \frac{3}{D_p^2 \left(\omega_p \mp \omega_s\right)^2} \qquad (22)$$

the minus sign accounts for co-propagation and plus sign for counter-propagation. Since only the real part of the $K$ function (18) will contribute to the gain, we have:

$$Re(K) = \cos\left(\frac{\Delta k\, L}{2}\right) \frac{\sin\left(\Delta k/2L\right)}{\Delta k L/2} = \frac{\sin\left(2\pi(L/l_p)\right)}{2\pi(L/l_p)} \qquad (23)$$

where the polarization decoupling length, related to the difference in the propagation constants $\Delta k = 2\pi/l_p$, has been explicitly used.

In this way, the gain coefficient (21) can be expressed in its polarization-dependent form:

$$g_{R_b} = \frac{Y P_0}{2} \left(1 \pm \frac{\sin\left(2\pi L/l_p\right)}{\left(2\pi L/l_p\right)}\right) \qquad (24)$$

We will have three scenarios:

- If $L \ll l_p \Rightarrow Re(K) \simeq 1$ so $g_{R_b} = Y P_0 = g_{R,max}^{PDG}$ ó $g_{R_b} = 0 = g_{R,min}^{PDG}$.

  In this case, there will be a strong dependence of the gain with polarization, either with parallel polarization and maximum value, or with orthogonal polarization and minimum value.
- If $L \gg l_p \Rightarrow Re(K) \simeq 0$, so $g_{R_b} = Y P_0/2$.

  Here, the gain coefficient for each main polarization axis becomes the same and independent of the input pump and signal polarization states. Its value turns out to be the average at $1/2$ of the maximum value in parallel polarization.
- For distances $L$ of the order of length $l_p$, the $g_{R_b}$ factor will fluctuate between the maximum and minimum of gain.

Once the expression for the gain coefficient (24) is obtained, to model our system with random birefringence, a succession of small equal-length fiber sections, with constant linear birefringence, that are coupled to each other, is considered [26]; we suppose also no correlation between adjacent

pieces. According to the previous hypothesis, we define the effective factor equivalent associated with the dependence of the gain with the polarization:

$$K_{eff}^{PDG} \equiv \left[\frac{1 \pm Re(K)}{2}\right]^{-1} = \frac{2}{1 \pm \dfrac{sin(2\pi(L_j/l_p))}{2\pi(L_j/l_p)}}, \qquad j = 1, 2, ..., N, \tag{25}$$

where $N$ is the number of small birefringent elements, the index $j$ corresponds to $j-$th element and $L_j = (j-1)\delta L$. Its graphical representation appears in Figure 1, which depicts the average value $1/K_{eff} \to 1/2$.

We can thus identify, in the propagation Equations (12) and (13), the parallel an orthogonal components of the gain and using this effective coefficient (25) for the gain average, we obtain:

$$\frac{g_{R\parallel}(\Omega_R)}{2} P_x(z) A_{s,x}(z,t) + \frac{g_{R\perp}(\Omega_R)}{2} P_y(z) A_{s,x}(z,t) \simeq \frac{g_R(\Omega_R)}{2K_{eff}^{PDG}(z)} \frac{P_p(z)}{2} A_{s,x}(z,t) \tag{26}$$

$$\frac{g_{R\perp}(\Omega_R)}{2} P_x(z) A_{s,y}(z,t) + \frac{g_{R\parallel}(\Omega_R)}{2} P_y(z) A_{s,y}(z,t) \simeq \frac{g_R(\Omega_R)}{2K_{eff}^{PDG}(z)} \frac{P_p(z)}{2} A_{s,y}(z,t) \tag{27}$$

where the total gain parameter modulated by the effective coefficient comes from both co-polarized and orthogonal polarized components signal–pump interaction:

$$g_R \approx g_{R\parallel} + g_{R\perp} \tag{28}$$

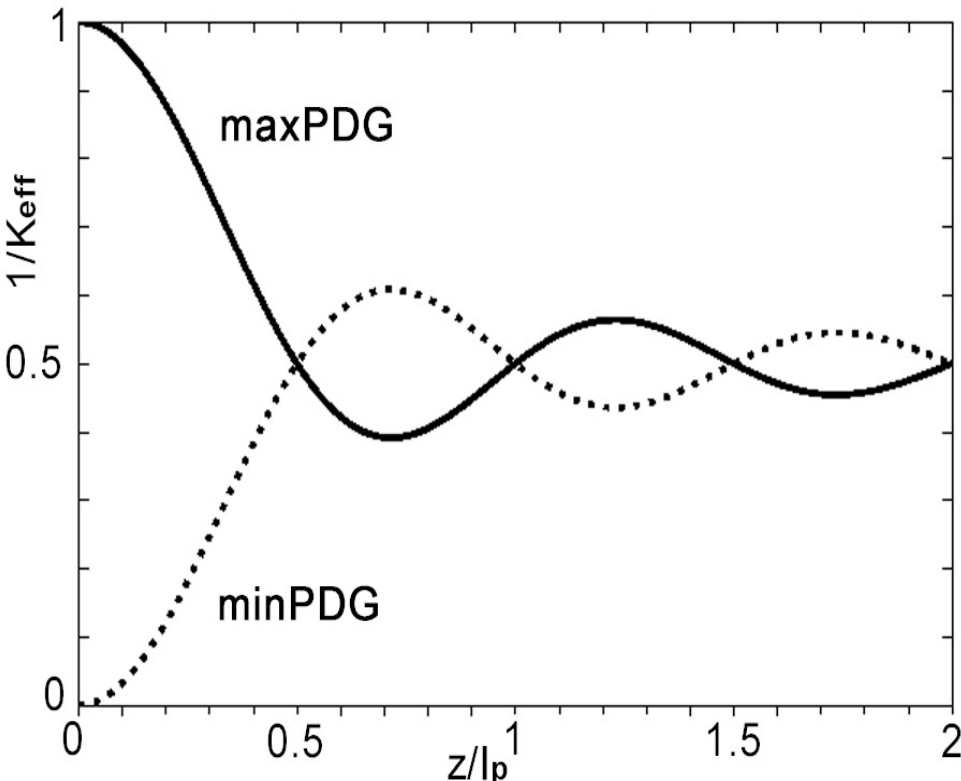

**Figure 1.** $K_{eff}^{PDG}$ vs. the normalized distance $z/l_p$.

*2.3. Averaging Vector Propagation Equations with Effective Raman Gain Term*

We used the Equations (12) and (13) with $B = 2/3$ to describe the propagation of polarization components in the pulse, so both components are coupled to each other in every fiber section via the

Kerr nonlinearity and to the pump via the gain term associated with the SRS effect. The effective coefficient $K_{eff}^{\text{PDG}}$ (25) takes into account the relationship between the Raman gain and the polarization. Without loss of generality, we can assume for a fixed value $z$ that the the coupling conditions are antisymmetric for the polarization components, i.e., when a component gets the maximum value of $K_{eff}^{\text{PDG}}$ the other one gets a minimum. This condition allow us to rewrite the coupled equations for the components (12) and (13):

$$\frac{\partial A_{s,x}}{\partial z} + \beta_{1s,x}\frac{\partial A_{s,x}}{\partial t} + \frac{i}{2}\beta_{2s}\frac{\partial^2 A_{s,x}}{\partial t^2} + \frac{\alpha}{2}A_{s,x} =$$
$$= i\gamma_s\left(|A_{s,x}|^2 + \frac{2}{3}|A_{s,y}|^2\right) + \frac{g_R(\Omega_R)}{2K_{eff}^{\text{PDG(max)}}(z)}\frac{P_p(z)}{2}A_{s,x}(z,t) \tag{29}$$

$$\frac{\partial A_{s,y}}{\partial z} + \beta_{1s,y}\frac{\partial A_{s,y}}{\partial t} + \frac{i}{2}\beta_{2s}\frac{\partial^2 A_{s,y}}{\partial t^2} + \frac{\alpha}{2}A_{s,y} =$$
$$= i\gamma_s\left(|A_{s,y}|^2 + \frac{2}{3}|A_{s,x}|^2\right) + \frac{g_R(\Omega_R)}{2K_{eff}^{\text{PDG(min)}}(z)}\frac{P_p(z)}{2}A_{s,y}(z,t) \tag{30}$$

where $K_{eff}^{\text{PDG(max)}}$ and $K_{eff}^{\text{PDG(min)}}$ are related, respectively, to the positive and negative signs in (25) and this evolution against the normalized distance is shown in Figure 1. Merging the loss term with the effective gain term $\alpha_s^{eff}(z)$, we get:

$$\alpha_s^{eff}(z)|_{min}^{max} = \alpha - \frac{g_R(\Omega_R)}{K_{eff}^{\text{PDG}_{min}^{max}}(z)}\frac{P_p(z)}{2} \tag{31}$$

and in order to simplify the equations for the polarization components, using the previously defined $\alpha_s^{eff}(z)|_{min}^{max}$ factor, these become:

$$\frac{\partial A_{s,x}}{\partial z} + \beta_{1s,x}\frac{\partial A_{s,x}}{\partial t} + \frac{i}{2}\beta_{2s}\frac{\partial^2 A_{s,x}}{\partial t^2} + \frac{\alpha_s^{eff}(z)|_{max}}{2}A_{s,x} =$$
$$= i\gamma_s\left(|A_{s,x}|^2 + \frac{2}{3}|A_{s,y}|^2\right) \tag{32}$$

$$\frac{\partial A_{s,y}}{\partial z} + \beta_{1s,y}\frac{\partial A_{s,y}}{\partial t} + \frac{i}{2}\beta_{2s}\frac{\partial^2 A_{s,y}}{\partial t^2} + \frac{\alpha_s^{eff}(z)|_{min}}{2}A_{s,y} =$$
$$= i\gamma_s\left(|A_{s,y}|^2 + \frac{2}{3}|A_{s,x}|^2\right) \tag{33}$$

From this point, we will transform the Equations (32) and (33) in their dimensionless form using:

$$u = \frac{A_{s,x}}{\sqrt{P_{s0}}}$$

$$v = \frac{A_{s,y}}{\sqrt{P_{s0}}}$$

where the peak power is:

$$P_{s0} = \sqrt{|A_{s,x}(0,0)|^2 + |A_{s,y}(0,0)|^2}$$

In a reference system moving along with the pulse components, we define the group velocity dispersion factor as:

$$\delta_s = \frac{\beta_{1s,x} - \beta_{1s,y}}{2} = \frac{1}{2}\left(\frac{n_{sx} - n_{sy}}{c}\right) = \frac{\Delta n}{2c} \tag{34}$$

The new temporal variable will be:

$$T = t - z\bar{\beta}_1$$

with $\bar{\beta}_1 = 1/2(\beta_{1s,x} + \beta_{1s,y})$. The new dimensionless variables:

$$\xi = \frac{z}{L_D}$$

$$\tau = \frac{T}{T_0}$$

with $T_0$ being the temporal width at the starting point and $L_D = T_0/\beta_{2s}$ the dispersion length.

The Equations (32) and (33), for the anomalous dispersion regime, after this change of variables appear as follows:

$$\frac{\partial u}{\partial \xi} + \delta\frac{\partial u}{\partial \tau} - \frac{i}{2}\frac{\partial^2 u}{\partial \tau^2} + \frac{\Gamma_s^{eff}|_{max}(\xi, \Omega_R)}{2}u = iN^2\left(|u|^2 + \frac{2}{3}|v|^2\right) \tag{35}$$

$$\frac{\partial v}{\partial \xi} - \delta\frac{\partial v}{\partial \tau} - \frac{i}{2}\frac{\partial^2 v}{\partial \tau^2} + \frac{\Gamma_s^{eff}|_{min}(\xi, \Omega_R)}{2}v = iN^2\left(|v|^2 + \frac{2}{3}|u|^2\right) \tag{36}$$

with:

$$N^2 = \gamma L_D P_{s0}$$

$$\Gamma_s^{eff}|_{min}^{max}(\xi, \Omega_R) = L_D\alpha_s^{eff}(\xi, \Omega_R)|_{min}^{max}$$

Due to the existence of the random orientation for the birefringence in our model, it is mandatory to make an additional transformation for the polarization amplitude variables, according to an aleatory rotation through a Poincaré sphere [27]:

$$\begin{bmatrix} u' \\ v' \end{bmatrix} = \begin{bmatrix} \cos\theta & \sin\theta e^{i\phi} \\ -\sin\theta e^{-i\phi} & \cos\theta \end{bmatrix}\begin{bmatrix} u \\ v \end{bmatrix} \tag{37}$$

the Equation (37) can be seen as an arbitrary rotation for the angles $2\theta$ and $\phi$ [28].

Doing the average over all the possible orientations and considering a pulse in the range of picoseconds, we obtain a version of the Manakov-PMD equations for the transformed components of the polarization in the distributed Raman gain regime:

$$\frac{\partial u'}{\partial \xi} - \frac{i}{2}\frac{\partial^2 u'}{\partial \tau^2} + \frac{\Gamma_s^{eff}|_{max}(\xi, \Omega_R)}{2}u' = iN^2\left(|u'|^2 + |v'|^2\right) \tag{38}$$

$$\frac{\partial v'}{\partial \xi} - \frac{i}{2}\frac{\partial^2 v'}{\partial \tau^2} + \frac{\Gamma_s^{eff}|_{min}(\xi, \Omega_R)}{2}v' = iN^2\left(|v'|^2 + |u'|^2\right) \tag{39}$$

The prior couple of equations forms the model that we are going to simulate for the study of our system with DRA in a fiber optic with random birefringence.

## 3. Simulation and Results

This section has two subsections. In the first one the optimal initial conditions, for the pulse launch, are calculated via the Method of the Moments and in the final part, we run a set of simulations to determine the influence of the distributed gain in the pulse propagation using the SSFM.

### 3.1. Obtaining Optimal Launch Conditions. Method of the Moments

Figure 2 shows the proposed optical communication system, based on a constant dispersion fiber, which works with DRA in a CW counter-pump regime. We assume that the propagation equations

are verified by the low variation envelopes of the electric field polarization components (38) and (39). These equations also model the losses present in the fiber and the SRS gain process induced by the pumping stations periodically located at $L_A$.

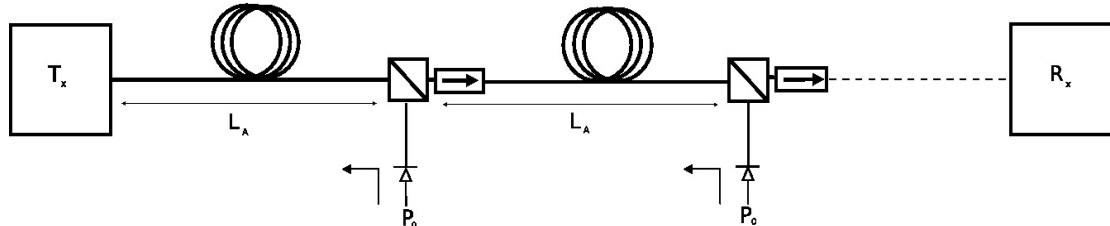

**Figure 2.** Scheme for a Raman amplification system in counter-propagation. $L_A$ accounts for the distance between pump stations.

We assume also the following conditions:

- The CW pumping model introduces a phase constant in the polarization components as a consequence of the XPM effect, but that constant has no impact on the overall study of the soliton propagation [29].
- The pumping power at each amplification stage will follow an exponential decay for which the effective length is given by:

$$L_{eff} = \frac{1 - \exp(-\alpha_p L_A)}{\alpha_p} \tag{40}$$

- Insulators are placed at each pump station, ensuring the performance of the system by avoiding interactions between consecutive pumping sections.

In our system, we can make use of the Method of the Moments to study the evolution of the optical pulses as they propagate through an optical fiber to determine the optimal launch conditions using the chirp and the peak power as parameters. First of all, to start with the method we define three moments:

1. The zero-order moment considering each pulse as a particle with energy:

$$E(\xi) = \int_{-\infty}^{+\infty} \left( |u(\xi, \tau)|^2 + |v(\xi, \tau)|^2 \right) d\tau \tag{41}$$

2. The first-order moment associated wit the central temporal position $T_m$:

$$\tau_m = \frac{1}{E} \int_{-\infty}^{+\infty} \tau \left( |u(\xi, \tau)|^2 + |v(\xi, \tau)|^2 \right) d\tau \tag{42}$$

3. The Root Mean Square (RMS) is given by the second-order moment:

$$\sigma^2 = \frac{1}{E} \int_{-\infty}^{\infty} (\tau - \tau_m)^2 \left( |u(\xi, \tau)|^2 + |v(\xi, \tau)|^2 \right) d\tau \tag{43}$$

the exact width is related to the RMS width through a constant factor that depends on the functional form of each pulse [30].

One more useful moment to introduce is the one associated with the chirp, which is defined by means of the polarization components $u$ and $v$:

$$\widetilde{C} = \frac{i}{2E} \int_{-\infty}^{+\infty} (\tau - \tau_m) \left( u^* \frac{\partial u}{\partial \tau} - u \frac{\partial u^*}{\partial \tau} \right) d\tau = \frac{i}{2E} \int_{-\infty}^{+\infty} (\tau - \tau_m) \left( v^* \frac{\partial v}{\partial \tau} - v \frac{\partial v^*}{\partial \tau} \right) d\tau \tag{44}$$

These moments will describe the evolution of the pulse parameters following a set of equations depending on the functional form of the pulse. In our study, with SRS counter-propagation at each

pump stage and despite of the possible variations of width and peak power, we will use the hyperbolic secant as the functional form of the solitons. Thus, we can consider each component as given by [31]:

$$u(\xi, \tau) = a_u(\xi) \text{sech}\left(\frac{\tau - \tau_m}{\eta(\xi)}\right) \exp\left[\frac{i\phi - iC(\xi)(\tau - \tau_m)^2}{2\eta^2(\xi)}\right] \tag{45}$$

$$v(\xi, \tau) = a_v(\xi) \text{sech}\left(\frac{\tau - \tau_m}{\eta(\xi)}\right) \exp\left[\frac{i\phi - iC(\xi)(\tau - \tau_m)^2}{2\eta^2(\xi)}\right] \tag{46}$$

where the phase factor $\phi$ does not depend on time or the pulse central position $\tau_m$ on $z$.

The width $\eta$ and chirp $C$ parameters shown in (45) and (46) are related with the RMS of the temporal width and $\widetilde{C}$ momentum through $\eta^2 = (12/\pi^2)\sigma^2$ and $C = (12/\pi^2)\widetilde{C}$ in the hyperbolic secant case considered.

As we can see, all the parameters represent local values and change with the axial direction of the fiber. If we introduce (45) and (46) in the non linear propagation equations with Raman amplification term (38) and (39), we find the pulse parameters ruled out by this system of coupled equations:

$$\frac{dE}{d\xi} = -\Gamma_s^{eff}|_{min}^{max}(\xi, \Omega_R)E \tag{47}$$

$$\frac{d\eta}{d\xi} = \frac{\beta_2 C}{\eta} \tag{48}$$

$$\frac{dC}{d\xi} = \left(\frac{4}{\pi^2} + C^2\right)\frac{\beta_2}{\eta^2} + \frac{2\gamma E}{\pi^2 \eta} \tag{49}$$

These equations are similar to the ones obtained using the variational method proposed in [32]. In our case, we have included the term that balance losses and gain in the system $\Gamma_s^{eff}|_{min}^{max}(\xi, \Omega_R)$.

Now, we apply this normalization:

$$e(\xi) = E/E_0 \qquad W = \eta/T_0 \tag{50}$$

with $E_0 = \int_{-\infty}^{+\infty}(|u(0, T)|^2 + |v(0, T)|^2)d\tau$ being the energy of each soliton at $\xi = 0$ and $T_0$ as the initial width having the same value for both polarization components and the dispersion length given by $L_D = T_0^2/\beta_2$.

Considering the anomalous propagation regime ($\beta_2 < 0$) along with (50), the moment equations for the energy (47), perturbed soliton width (48) and chirp (49) can be written as:

$$\frac{de}{d\xi} = -\Gamma_s^{eff}|_{min}^{max}(\xi, \Omega_R)e \tag{51}$$

$$\frac{dW}{d\xi} = -z_A \frac{C}{W} \tag{52}$$

$$\frac{dC}{d\xi} = \frac{4}{\pi^2} z_A N^2 \frac{e}{W} - \left(\frac{4}{\pi^2} + C^2\right)\frac{z_A}{W^2} \tag{53}$$

where $z_A = L_A/L_D$ is the normalized length between consecutive pump stations and $N^2 = \gamma P_0 L_D$ the normalized initial peak power (soliton order).

The basic idea is to obtain a periodic solution of these parameters that allows the perturbed soliton to recover its initial values (except for a phase factor) after each normalized amplification stage. We are interested in boundaries that assure the launch parameters periodicity:

$$C(0) = C(1) \qquad W(0) = W(1) = 1 \tag{54}$$

To solve numerically the generic form of our system of coupled Equations (51)–(53), we employ an algorithm based on the *Runge–Kutta–Fehlberg* (R–K–F) method with *Cash-Karp* 6th-order coefficients [33]. We have also fixed the maximum *Raman* effective gain value at $g_R^{eff}|^{max}(\Omega_R) = 1,4$ (Km·W)$^{-1}$ with signal and pumping wavelengths of $\lambda_s$ = 1550 nm and $\lambda_p$ = 1450 nm and an effective area of $A_{eff}^{ps} = 50 \times 10^{-12}$ m$^2$ and with $\alpha_p = \alpha_s = \alpha = 0.21$ dB/Km. The pump–station length was taken periodically at $L_A$ = 40 km. The gain will be $G = \exp(\alpha L_A) = 10$ ($\alpha\, L_A = 2.3$) in each amplification stage.

We have carried out simulations starting from the analytic expressions of normalized peak power and initial chirp with $z_A \ll 1$ [12], we varied the dispersion length $L_D$, and thus $z_A$, until we found the $N^2$ and $C_0$ values that assure the boundaries given by (54) at each $z_A$. Using a range of values from $z_A = 0$ to $z_A = 5$, we have obtained the optimal launch conditions, i.e., normalized input peak power and soliton chirping, as a function of normalized pump–station spacing $z_A$. The results are shown in Figure 3.

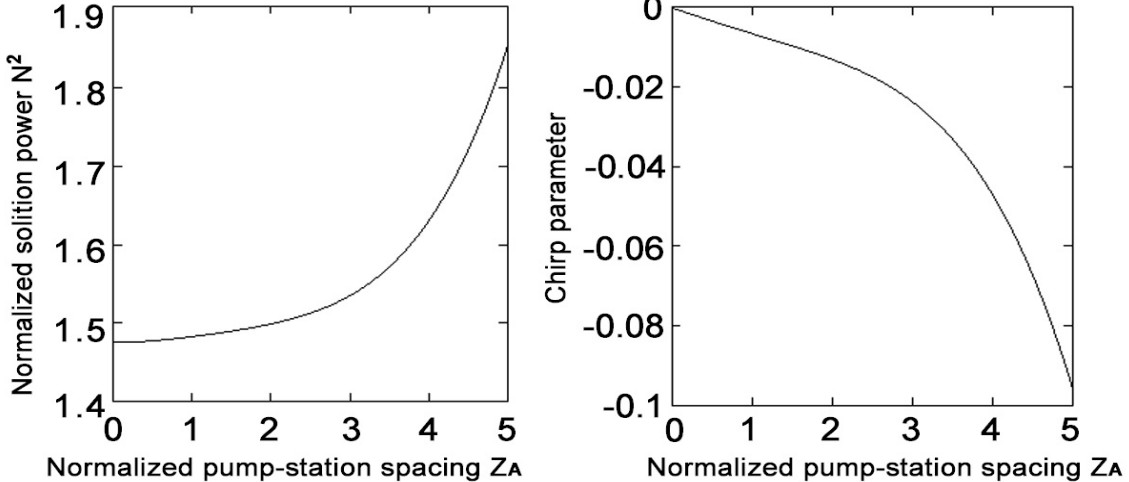

**Figure 3.** Normalized input peak power and chirp of soliton as a function of normalized pump–station spacing $z_A$ numerically obtained from Method of Moments when consecutive pumping stages are separated by 40 km.

We can point out two main conclusions: the chirp parameter, which displays a highly sensitive behaviour as can be seen in the right side of Figure 3, allows a greater spacing between the pumping stations, even exceeding the dispersion length; on the other hand, the control in the pumping power is the most important factor to ensure the required periodicity conditions. This is due to the fact that in such systems the soliton power evolves, regenerating itself during the journey.

### 3.2. Effect of the Distributed Gain on the Energy and Width of the Propagating Vector Pulses

To validate the previous results, we try to regenerate the pulses with the same injected power. At the beginning of the fiber $z = 0$, we launch vectorial chirped solitons with a wavelength of $\lambda_s = 1550$ nm, at this point the initial conditions have the functional form:

$$u(\xi = 0, \tau) = A\cos\theta \sec h(\tau) \exp(-iC\tau^2/2) \tag{55}$$

$$v(\xi = 0, \tau) = A\sin\theta \sec h(\tau) \exp(-iC\tau^2/2) \tag{56}$$

where $A$ is the amplitude related with the soliton order, $C$ is the lineal chirp parameter that takes into account for the initial phase in both components of the pulse and $\theta$ determines the relative intensity of every component. The condition:

$$P_0 = |u(\xi = 0, \tau = 0)|^2 + |v(\xi = 0, \tau = 0)|^2 = |A|^2(\cos^2\theta + \sin^2\theta) = |A|^2 \tag{57}$$

is also accomplished.

Our hypotheses for the simulation are:

- The study of the propagation characteristics for the chirped vectorial solitons (CVS) has been performed using a predictor–corrector numerical model with an SSFM [34] to resolve the propagation Equations (38) and (39).
- The random birefringence effect on the gain is included in the effective coefficient $K_{eff}^{\text{PDG}}$, through the decoupling length $l_d$ between the polarization states of the pump and the signal, using Equation (22). In the conventional scenario for a monomode fiber, the values of the parameter $D_p$ are included in the range $0, 1 - 1 \, ps/\sqrt{km}$, this election turns into a $l_p$ distance not bigger than a kilometer.
- In that way, the amplification effective length for every section is bigger than the $l_p$, i.e., $L_{eff} > l_p$, assuring the complete decoupling between the polarization states of the signal and pump in the propagation process, achieving the same amplification for both signal components and with the same average value. In our model, this effect is clearly bring out since the effective coefficient $1/K_{eff}^{\text{PDG}}$ reaches the 1/2 value before the pulse propagation in each one of the amplification zones $L_A$.
- The simulations have been carried out in soliton systems with $T_0 \sim$ 5–10 ps width in the anomalous dispersion regime, which allow an effective transmission rate of 10–40 Gb/s. Our study is centered on the effect of initial control of the phase via the lineal chirp and the optimal peak power, obtained according the Method of Moments, in the stability of the vectorial solitons with Raman gain. For completeness, different values for the parameter $z_A = L_A/L_D$ have been used.

To start the simulations, we will check the complete regeneration of the energy value, in a system with equal values of the amplitude in both components, i.e., $\theta = \pi/4$. We will analyze the effect of the pump distributed gain in counter-propagation on the total energy.

Using the adjusted pump power $P_p(L_A)$ for $L_A = 40$ km, we can calculate the energy (41):

$$E(z) = \int_{-T_b/2}^{+T_b/2} \left( |u(z,T)|^2 + |v(z,T)|^2 \right) dT \tag{58}$$

where $T_b$ is the temporal width of the bit slot where the vectorial soliton is placed.

In Figure 4, the evolution of the energy is represented in an amplification stage with $z_A = 2$ and optimal pump and chirp conditions according to Figure 3.

As we can see, the effect of the losses during the first part of the propagation makes the energy decrease but the stimulated Raman scattering, when circulating in counter-propagation, regenerates completely the energy of the soliton at the end of the amplification stage, as expected due to the periodic conditions implemented in the Method of the Moments.

In the same way, we have tested the stability of the propagation. To do that, we have performed a scan in the value of the chirp parameter to calculate the widening average, measured as standard deviation ratio for every vectorial pulse at the end of an amplification stage of $L_A = 40$ km. The results for these simulations are shown in Figure 5 with $Z_A = 0.5$ and Figure 6 with $Z_A = 2$.

It can be seen how the optimal chirp value obtained allows us to keep the ratio $\sigma/\sigma_0$ equal to 1, if we also use the optimal peak power for every $z_A$ value, as appears in Figure 3. The normalized width for different values of the initial chirp using the optimal peak power and values 10% over (superior line) or under (inferior line) the optimal is also shown in Figures 5 and 6.

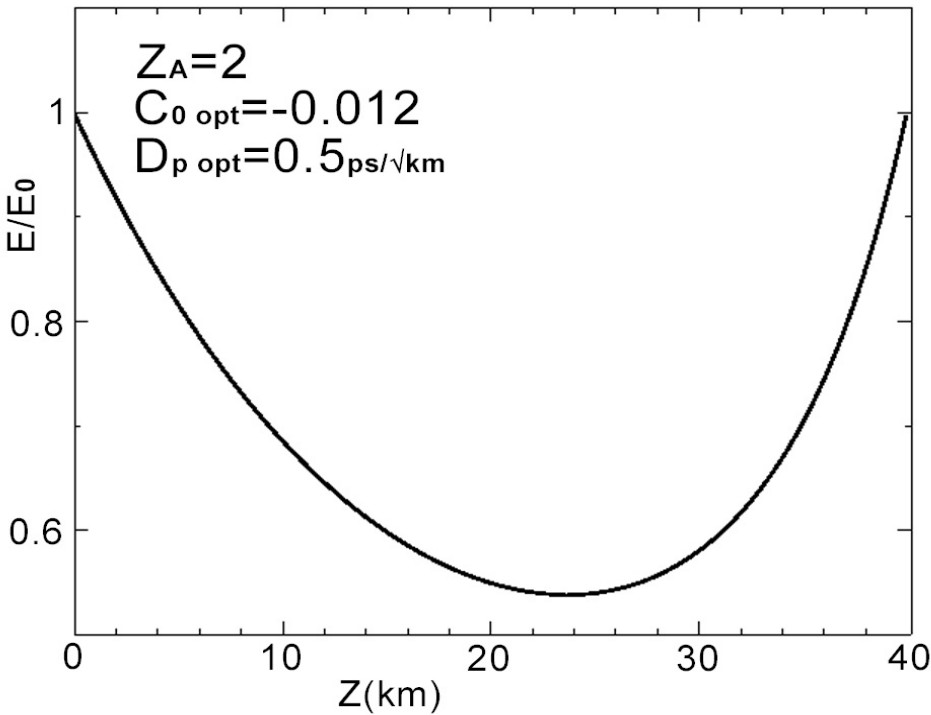

**Figure 4.** Evolution of the energy for the solitonic vectorial pulse with $z_A = 2$ in one amplification stage in optimal pump conditions $P_{0opt}$, shared between the polarization components according to (55) and (56), with $\theta = \pi/4$ and optimal chirp. The value for the polarization mode dispersion (*PMD*) parameter is $D_p = 0.5$ ps $\cdot$ km$^{-1/2}$.

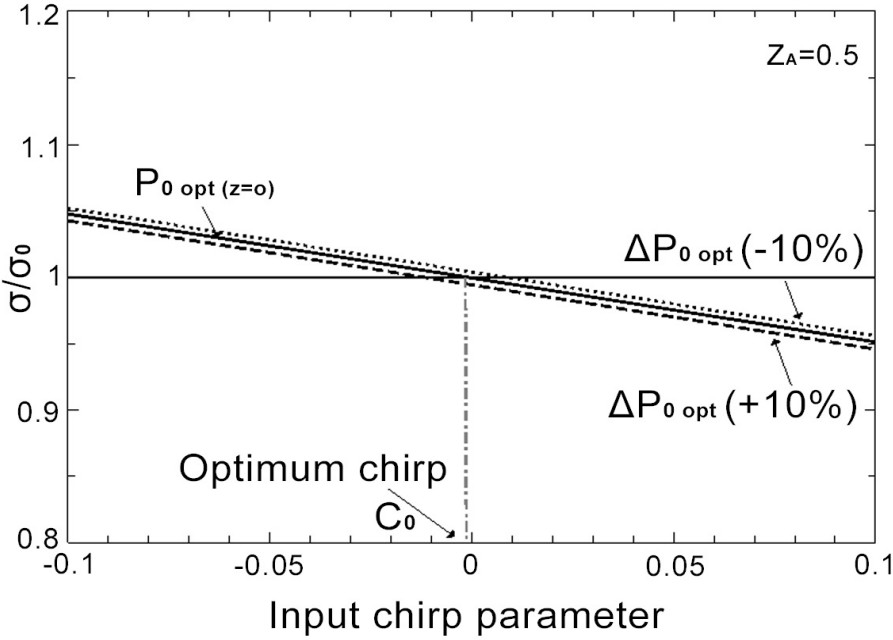

**Figure 5.** Change in the widening factor through an amplification stage for the vectorial soliton with $z_A = 0.5$ vs. the initial chirp, for different peak powers in $z = 0$. We assume a PMD value of $D_p = 0.5$ ps $\cdot$ km$^{-1/2}$.

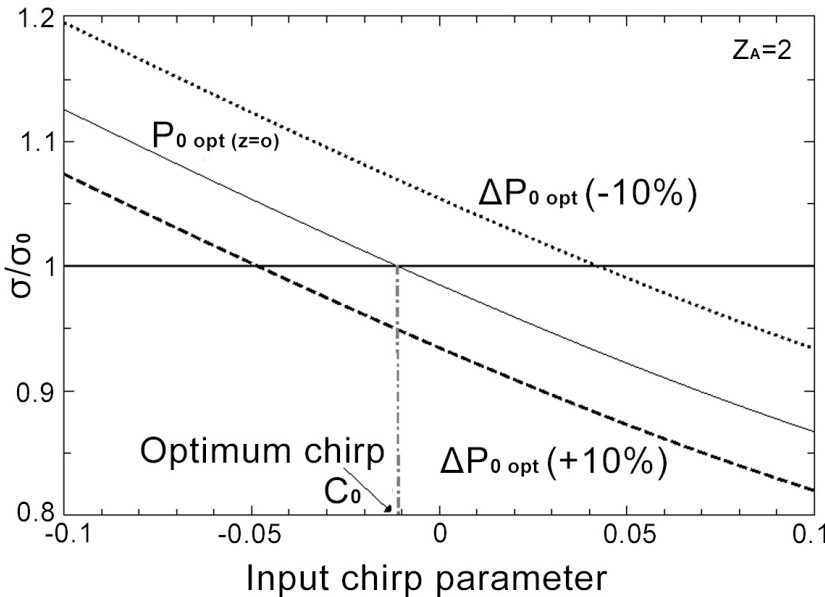

**Figure 6.** Change in the widening factor through an amplification stage for the vectorial soliton with $z_A = 2.0$ vs. the initial chirp, for different peak powers in $z = 0$. We assume a PMD value of $D_p = 0.5$ ps · km$^{-1/2}$ .

We can appreciate, comparing Figures 5 and 6, how the separation between the lines corresponding to the non-optimal peak power values is greater in the case of $z_A > 1$ (Figure 6) due to the relevance of the dispersive effects. In any case, we can assure the periodicity for the pulse after every stage, so we can conclude that in optimal conditions the propagation of the vectorial solitons is stable in DRA systems.

Finally, to display these periodicity conditions in a more clear way, we show in Figure 7 the evolution of the widening parameter for a vectorial solitonic pulse in an amplification stage with a value of $z_A = 2$ with optimal chirp. It can be seen how a correct election in the peak power ensures the periodicity in the $\sigma/\sigma_0$ parameter, so we can consider this result as a validation of the Method of the Moments for the study of amplification systems with random birefringence.

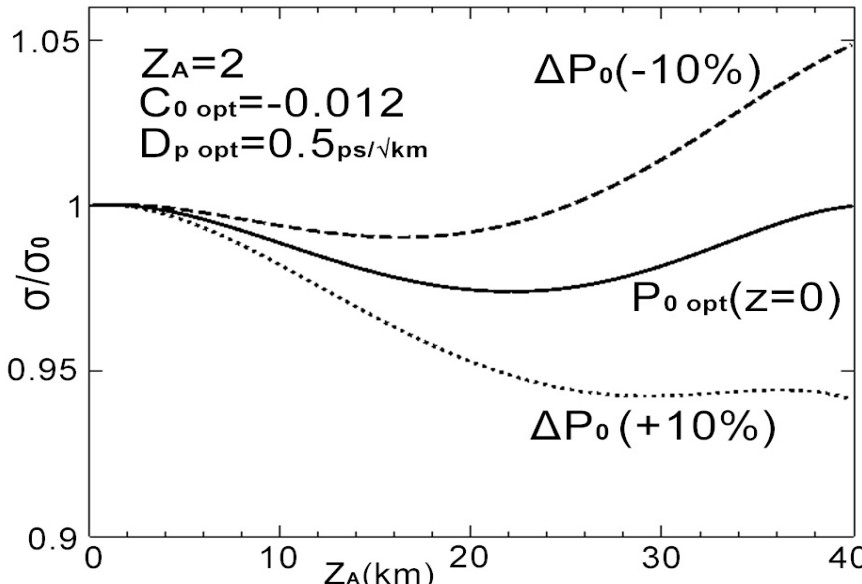

**Figure 7.** Evolution of the normalized widening with $z_A = 2$ in an amplification stage with different initial power peaks: optimal power (—), 10% over (· · ·) and 10% under(− − −), in all the three cases shared between components according to (55) and (56) with $\theta = \pi/4$ and optimal chirp. The value for the PMD parameter is $D_p = 0.5$ ps · km$^{-1/2}$.

## 4. Conclusions

The vector description that we have made of solitons propagation in an optical communication system with the use of backward-distributed Raman amplification in fibers with random birefringence allows us to ensure the stability of these pulses. For distances commonly used in amplification systems greater than decoupling and polarization beat, the numerical resolution of propagation equations confirms how non-linearity and Raman gain interaction between the pumping and signal polarization states bring a homogeneous amplification of them, minimizing the dependence of gain effect with polarization. It also validates the predictions of the Method of Moments, which provides the optimal launch conditions to maintain, on average, the integrity of these pulses during their propagation over long fiber runs, after successive stages of distributed Raman amplification.

**Author Contributions:** Conceptualization, A.O.-M.; Investigation, A.O.-M.; Methodology, A.O.-M.; Project administration, A.D.; Software, A.O.-M.; Supervision, P.R., A.D.-S. and A.D.; Validation, D.M.-M.; Visualization, P.R. and A.D.-S.; Writing—original draft, A.O.-M.; Writing—review & editing, P.R., A.D.-S., D.M.-M. and A.D. All authors have read and agreed to the published version of the manuscript.

**Funding:** This research received no external funding. We would like to thank the Physics Department of the University of Córdoba for its supporting during researching process.

**Acknowledgments:** The authors appreciate the support of C. Quesada Padilla with the grammatical corrections of the final document.

**Conflicts of Interest:** The authors declare no conflict of interest.

## Abbreviations

The following abbreviations are used in this manuscript:

| | |
|---|---|
| CVS | Chirped Vectorial Soliton |
| CW | Continuous Wave |
| DRA | Distributed Raman Amplification |
| FWM | Four Wave Mixture |
| PMD | Polarization Mode Dispersion |
| RFA | Raman Fiber Amplifier |
| SPM | Self Phase Modulation |
| SSFM | Split-Step Fourier Method |
| SRS | Stimulated Raman Scattering |
| XPM | Cross-Phase Modulation |

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
