# Peer review of "Method of Moments Optimization of Distributed Raman Amplification in Fibers with Randomly Variying Birefringence"

_photonics, doi:10.3390/photonics7040086_

Round 1

Reviewer 1 Report

Review of the manuscript Photonics-944583 “Improving pulse propagation in fiber optics distributed Raman amplification with randomly varying birefringence”

By A. Ortiz-Mora,   P. Rodriguez, A. Diaz-Soriano, D. Martinez-Muoz and A. Dengra.

The authors developed in the proposed article a novel vectorial model of the vectorial soliton pulses in monomode fiber optics taking into account the influence of the Raman amplification. The topic of the article is important for the improvement of the optical communication system performance. The authors created a generalized theoretical model of optical signal propagation in a fiber taking into account the optical nonlinearity, dispersion effects, and polarization effects. The theoretical level of the article is high. The authors derived the truncated equations for the optical field components and the nonlinear polarization in the optical fiber. They presented some analytical expressions and the numerical simulation results. The proposed article is clearly written and well organized. The results are interesting for the researchers and engineers occupied in the field of the optical communications in general and in the nonlinear fiber optics in particular. The article may be published in the Journal “Photonics” in a present form.

I recommend the correction of the misprints in equations (18), (23)-(25):

It should be .

Author Response

   Thank you very much for your kind words. As you recommended, we have fixed the typos in the equations that you pointed out in your review. Thanks in regard.

Reviewer 2 Report

In the paper “Improving pulse propagation in fiber optics distributed Raman amplification with randomly varying birefringence” the authors describe computer simulations to model the propagation of vectorial solitons for distributed Raman amplification in fibers with randomly distributed birefringence. They calculate a polarization dependent Raman effective gain coefficient; they also study the optimization of the initial conditions for the pulse and the influence of dispersion.

I find the paper relevant and worth of publication since it covers a facet that has not been modeled before. I do not have any comments from the technical point of view, I believe the model presented is sound and improves on the existing understanding of soliton propagation. The paper, in general, is well written; my few observations follow:

  1. I suggest to revise the title since it is somewhat confusing. I believe this is due mainly to the English usage.
  2. The paper is somewhat long, mainly because several equations are written twice (once for each polarization). Using some convention for the subindex notation or explicitly indicating where a “permutation” of polarizations lead to similar equations could shorten the manuscript.
  3. There are a few typos and there are several parts where the right margin is not flushed; although, obviously, this is not important for the content of the paper, sometimes it is distracting.
  4. In line 33, I think that what the authors propose is not “new hypothesis”, but rather “model assumptions and/or limitations”.
  5. In line 96, there is a mistake in the latex code that results in bet a instead of /beta.

In line 263 says that the chirp parameter can be seen in the left side of figure 3; while it is in the right side.

Author Response

Thank you very much for your in deep revision. We found it very useful and we want to make the next comments on that:

1. I suggest to revise the title since it is somewhat confusing. I believe this is due mainly to the English usage.

We have changed the title in order to make it less confusing and shorter.

2. The paper is somewhat long, mainly because several equations are written twice (once for each polarization). Using some convention for the subindex notation or explicitly indicating where a “permutation” of polarizations lead to similar equations could shorten the manuscript.

We agree that we could have chosen another notation system, but we wanted to make more explicit the equations in order to clarify the text.

3. There are a few typos and there are several parts where the right margin is not flushed; although, obviously, this is not important for the content of the paper, sometimes it is distracting.

We hope that the edition process correct this issue.

4. In line 33, I think that what the authors propose is not “new hypothesis”, but rather “model assumptions and/or limitations”.

We have changed the terms as you proposed.

5. In line 96, there is a mistake in the latex code that results in bet a instead of /beta.

We have corrected the mistake.

Thank you again.

Reviewer 3 Report

The authors develop a mathematical model for nonlinear optical pulse propagation in a single mode fiber in the anomalous dispersion regime with the distributed Raman amplification. The model takes into account the effect of random birefringence of the fiber on the Raman gain coefficient. The authors then reduce the system of coupled partial differential equations to obtain a set of three ordinary differential equations that approximates the evolutions of the pulse energy, width and chirp. This is then used to obtain the optimal launch conditions for an example optical communication system using the distributed Raman amplification system. They find the initial chirp and pump power are the critical parameters for optimizing the distributed Raman amplification based optical communication line. These findings are validated by numerically solving directly the coupled partial differential equations.

I find the study to be scientifically sound and interesting. I did not find any major pitfalls in their approach. I should mention, however, the use of the English language needs to be improved. For example, I see in several instances, multiple sentences are joined with commas in between. See e.g., Lines 45 - 48 in Page 2. There are many places where "y" is used instead of "and", and "sen" is used instead of "sin". Also, the authors need to be careful with the choice of the symbols. For example, in Eq. (48) tau is defined as the pulse width, but it has been used also as the moving time variable. Some notations are not defined. For example, "m" in Line 67 in Page 2, "x_i" in Line 242 in Page 12 appear out of nowhere.

I recommend the publication of this work after having addressed the above mentioned issues.

Author Response

Thanks for your kind words. We have replaced the grammatical errors mentioned in the review, an also changed the mistakes on the notation.